# The Phenotypic and Genetic Spectrum of Glycogen Storage Disease Type VI

**DOI:** 10.3390/genes12081205

**Published:** 2021-08-03

**Authors:** Sarah Catharina Grünert, Luciana Hannibal, Ute Spiekerkoetter

**Affiliations:** 1Department of General Paediatrics, Adolescent Medicine and Neonatology, Faculty of Medicine, Medical Centre-University of Freiburg, 79106 Freiburg, Germany; ute.spiekerkoetter@uniklinik-freiburg.de; 2Laboratory of Clinical Biochemistry and Metabolism, Department of General Paediatrics, Adolescent Medicine and Neonatology, Faculty of Medicine, Medical Centre-University of Freiburg, 79106 Freiburg, Germany; luciana.hannibal@uniklinik-freiburg.de

**Keywords:** glycogen storage disease, GSD, PYGL, hepatic cirrhosis, hepatic fibrosis, ketotic GSD, carbohydrate metabolism

## Abstract

Glycogen storage disease type VI (GSD VI) is an autosomal recessive disorder of glycogen metabolism due to mutations in the glycogen phosphorylase gene (*PYGL*), resulting in a deficiency of hepatic glycogen phosphorylase. We performed a systematic literature review in order to collect information on the clinical phenotypes and genotypes of all published GSD VI patients and to compare the data to those for GSD IX, a biochemically and clinically very similar disorder caused by a deficiency of phosphorylase kinase. A total of 63 genetically confirmed cases of GSD VI with clinical information were identified (median age: 5.3 years). The age at presentation ranged from 5 weeks to 38 years, with a median of 1.8 years. The main presenting symptoms were hepatomegaly and poor growth, while the most common laboratory findings at initial presentation comprised elevated activity of liver transaminases, hypertriglyceridemia, fasting hypoglycemia and postprandial hyperlactatemia. Liver biopsies (*n* = 37) showed an increased glycogen content in 89.2%, liver fibrosis in 32.4% and early liver cirrhosis in 10.8% of cases, respectively. No patient received a liver transplant, and one successful pregnancy was reported. Our review demonstrates that GSD VI is a disorder with broad clinical heterogeneity and a small number of patients with a severe phenotype and liver cirrhosis. Neither clinical nor laboratory findings allow for a differentiation between GSD VI and GSD IX. Early biochemical markers of disease severity or clear genotype phenotype correlations are missing. Given the overall benign and unspecific phenotype and the need for enzymatic or genetic analyses for confirmation of the diagnosis, GSD VI is likely underdiagnosed. With new treatment approaches in sight, early, pre-symptomatic diagnosis, especially with respect to hepatic cirrhosis, will become even more important.

## 1. Introduction

Glycogen storage disease type VI (GSD VI, OMIM #232700) is an inborn error of glycogen metabolism caused by biallelic mutations in the PYGL gene that result in a deficiency of hepatic glycogen phosphorylase (PYGL) [1,2]. This enzyme catalyzes the rate-limiting step of glycogen degradation. Glycogen phosphorylase has different tissue-specific isoforms including brain (PYGB), heart (PYGB), muscle (PYGM) and liver (PYGL) [3]. Among those, liver-specific PYGL is the only isoform that allows the rapid release of free glucose from glycogen into circulation, thus stabilizing blood glucose levels to provide energy to extrahepatic tissues [3]. 

GSD VI, also called Hers disease, was first reported by Henry-Gery Hers in 1959 [4]. Genetic confirmation of GSD VI disease became available only in 1998 [2]. The estimated incidence of GSD VI is approximately 1:65,000–1:85,000 live births [1]. Most patients present early in childhood. The clinical and laboratory findings comprise hepatomegaly, poor growth, short stature, ketotic hypoglycemia, elevated hepatic transaminases, hypertriglyceridemia and hypercholesterolemia [1,5]. 

There is a paucity of literature on GSD VI, and current knowledge on the natural history of the disease mainly derives from case reports or small case series. Especially, long-term outcome data on patients with GSD VI are scarce. Published findings suggest significant clinical heterogeneity, and although GSD VI is usually considered a relatively mild disorder [1], some severe cases with recurrent hypoglycemia, liver cirrhosis or developmental delay have been described [6,7,8].

Some individuals with GSD VI may not require any treatment; others significantly benefit from a high-protein diet (2–3 g/kg/day) with frequent small meals and supplementation of uncooked cornstarch to improve growth [1,5]. A reduction of total carbohydrates and especially simple sugars is recommended, to reduce glycogen storage in the liver [1]. 

Fernandes et al. recently published a review on all genetically proven cases of GSD type IX, a very similar disorder of glycogen metabolism due to phosphorylase kinase deficiency [9]. Phosphorylase kinase catalyzes the phosphorylation and thereby activation of glycogen phosphorylase as a crucial regulatory enzyme in glycogenolysis. The authors analyzed the data of 230 patients with different subtypes of GSD IX, with a special focus on age at diagnosis, presenting symptoms, laboratory findings, liver biopsy results, liver transplant and genetic findings [9]. GSD VI and IX are usually considered disorders that cannot be distinguished by clinical or laboratory findings other than by enzymatic or genetic analyses. With the aim of facilitating the distinction of GSD VI and GSD IX, this review integrates findings from all published cases of genetically proven GSD VI. In order to be able to compare our data on GSD VI to the data for GSD IX provided by Fernandes et al. [9], we chose a very similar approach of literature search and evaluation.

## 2. Methods

### 2.1. Literature Review

Information on the clinical phenotypes and genotypes of all published GSD VI patients was collected from a systematic literature search in PubMed in May 2021. The search was conducted using the following search terms to ensure maximal coverage of reported cases: “Glycogen storage disease type VI”, “GSD VI”, “glycogenosis type VI”, “glycogenosis type 6”, “Glycogen storage disease type 6”, “GSD 6, “Glycogen storage disease type six” and “PYGL”. 

Only patients with genetically confirmed GSD VI and relevant clinical information in the respective publication(s) were included in this study. With this approach, we identified a total of 63 individuals with symptomatic GSD VI. The list of publications included in this analysis is shown in Appendix A. In case of duplicate publications of single individuals, information was retrieved and merged from both publications. Reports on larger cohorts of patients, from which information on single patients could not be extracted as data, were only given as overviews on the total cohort and were excluded from analysis (i.e., [10,11]). Similarly, reports on genetic data without clinical information (i.e., [12]) and patient reports without genetic confirmation of the diagnosis were excluded (i.e., [13]).

### 2.2. Data Collection

All relevant clinical data were extracted from the collected literature. In order to compare findings from GSD VI patients with those recently published for GSD IX patients [9], the datasets were organized and evaluated according to Fernandes et al. [9]. Briefly, all the cases were evaluated and analyzed with a special focus on demographic information; the clinical course, including the age at diagnosis and presenting symptoms; biochemical laboratory, enzymatic and genetic data; liver biopsy findings; hepatic adenoma; transplantation; and pregnancies.

#### 2.2.1. Demographic and Genetic Information

The demographic information included the gender, age at diagnosis, presenting symptoms, affected siblings and *PYGL* genotype. 

#### 2.2.2. Hepatomegaly, Growth and Development

Patients with documented hepatomegaly, growth retardation, delayed growth or short stature were reported according to published criteria [9]. All the information on neurologic outcomes or neurologic symptoms was collected from the case reports.

#### 2.2.3. Biochemical Laboratory Data

The following laboratory data were collected: the presence or absence of fasting ketosis, hypertriglyceridemia, hypercholesterolemia, fasting hypoglycemia, hyperlactatemia, elevated transaminase activities and elevated creatine kinase activity. The results of enzyme activity studies were also evaluated. 

#### 2.2.4. Liver Biopsy, Hepatic Adenoma and Transplantation

Information on liver biopsies and the age at biopsy, increased glycogen storage, the presence of fibrosis and/or cirrhosis, liver transplantation, the age at transplantation and the presence of hepatic adenoma was collected. 

#### 2.2.5. Data Analysis

We performed descriptive statistics. To calculate the frequency of specific findings, only patients with available information for the specific item were included in the respective analysis. 

## 3. Results

### 3.1. Demographic Data

We identified 63 patients with genetically confirmed GSD VI for whom relevant clinical information was available in the literature. Thirty-three patients (52.4%) were male; 30 were female (47.6%). The patients derived from 56 unrelated families. In 15 families (26.8%), parental consanguinity was reported. The patient cohort included four pairs of siblings, thereof one pair of non-identical twins and one additional family with three affected children. All the patients were alive at the time of report. The median age at the time of the report was 5.3 years (*n* = 40; range: 1.4–41 years). 

Information on ethnic or geographic background was available for 47 families. Thirty-one-point-nine percent were Caucasian/of European origin, 36.2% Chinese, 8.5% Turkish, 6.4% Indian and 4.3% Algerian. The single patient cases were of Central African, Israeli Arab Bedouin, British Asian, Tunisian, Suriname-Hindustani and Mennonite origin (Figure 1).

### 3.2. Diagnosis and Initial Presentation

The age at presentation was reported for all the patients; however, in two cases, the information was not exact enough (“before 4 years of age” and “in early infancy”) to be included in the calculation. The age at presentation ranged from 5 weeks to 38 years, with a median of 1.8 years (*n* = 61). An overview on the age at initial presentation is given in Figure 2.

The main presenting symptoms were hepatomegaly and poor growth. In many cases, parents had recognized an enlarged abdomen months before the diagnosis was made. An overview on the presenting symptoms in this cohort of 63 patients at initial presentation is shown in Table 1. 

The most common laboratory findings at initial presentation comprised elevated activities of liver transaminases, hypertriglyceridemia, fasting hypoglycemia and postprandial hyperlactatemia. Although triglyceride levels were mostly only mildly elevated, triglyceride concentrations up to 1400 mg/dl were reported in single patients. The frequencies of laboratory abnormalities at presentation are displayed in Table 2.

### 3.3. Liver Biopsy

Liver biopsies were performed in 37 patients (median age at biopsy: 2.6 years; mean age at biopsy: 2.9 years; *n* = 16). Increased glycogen storage was detected in 33 patients (33/37; 89.2%), and liver fibrosis ranging from mild to marked was found in 12 patients (12/37; 32.4%). Liver cirrhosis was diagnosed in four patients (4/37; 10.8%). In all four, cirrhosis was found as early as at preschool age (age at biopsy: 1.4–6.1 years). All four were Chinese but had different genetic backgrounds. In one patient (1/37; 2.7%), mild steatosis was found.

### 3.4. Liver Adenoma

Adenoma formation was not reported in this patient cohort. One nine-year-old patient, reported by Aeppli et al., developed a focal nodular hyperplasia in liver segment III (diagnosis made by ultrasound and confirmed by liver biopsy) that showed only little progression and was under observation at the time of report [14]. 

### 3.5. Liver Transplantation

None of the patients received a liver transplant. 

### 3.6. Mutation Analysis and Enzyme Data

The results of the *PYGL* mutation analysis were available for all the patients. Thirty-one patients (31/63; 49.2%) were compound homozygous, while 30 patients (30/63; 47.6%) carried homozygous mutations. In one patient (1/63; 1.6%), only one variant could be detected, but the diagnosis was confirmed by the detection of reduced phosphorylase-A activity in liver and muscle samples. In another patient (1/63; 1.6%) with reduced phosphorylase activity in lymphocytes, two mutations were identified, but both were located on the maternal allele. A total of 63 PYGL variants were detected, including 36 missense mutations, seven stop mutations, 12 splice site variants, seven deletions and one insertion. 

Enzymatic analyses were performed in 29 patients and confirmed phosphorylase-A deficiency in either liver or blood cells in all cases.

### 3.7. Treatment

Information on treatment was available for 30 patients. In general, information on treatment was very limited. Twenty-eight out of 30 patients (93%) received cornstarch, and nine (9/30; 0.3%) were recommended a high-protein diet.

### 3.8. Long-term Outcomes and Complications 

As the median age at the time of report was 5.3 years in this patient cohort, information on the long-term outcomes is limited. Follow-up growth data were reported for 23 patients. Thereof, nine patients (39%) showed normal growth. In three patients with previously impaired growth, normalization was observed during further examinations (13.0%), seven patients (30.4%) showed improved growth and in four patients, no growth improvement was observed, with persistent short stature. One patient with normal growth at the time of diagnosis developed an atypical eating disorder and transiently lost weight but showed a catch-up growth to her previous percentiles thereafter. Two patients required a percutaneous endoscopic gastrostomy (PEG) tube. Hepatomegaly was persistent in 15 of 24 patients (62.5%), improved in four (4/24; 16.7%) and normalized in another five (5/24; 20.8%) patients. Transaminase activities were normalized in 18 of 24 patients (75.0%); decreased, but not fully normalized, at the time of the report in five patients (5/24; 20.8%); and remained unchanged in one patient (1/24; 4.2%). These data, however, need to be interpreted with caution, as the follow-up time of the patients was very variable and only five patients were older than 10 years.

Apart from liver-specific complications, other long-term complications were rare. In one patient, academic achievement was reported to be below average [7], but no other neurologic complications were observed. No patient developed symptomatic cardiomyopathy, but in one child, an increase in septum wall thickening (z-score: −3.2) and left ventricular posterior wall thickening was reported at the age of 4 years and 3 months [15].

### 3.9. Pregnancy

Only one pregnancy has been reported in a woman with GSD VI to date [16]. The 38-year-old patient became pregnant after fertility treatment, and apart from ovarian hyperstimulation syndrome, the pregnancy was uncomplicated and resulted in healthy offspring.

## 4. Discussion

GSD VI is usually considered a relatively mild disorder [1]; however, more severe cases with recurrent hypoglycemia, liver cirrhosis or developmental delay have been reported [6,7,8]. Phenotypic information on GSD VI is currently only available from published case reports and small case series. Here, we present the first comprehensive literature review on GSD VI including 63 genetically proven cases. The most common presenting symptoms were hepatomegaly and poor growth. The most common laboratory findings at initial presentation were elevated activities of liver transaminases, hypertriglyceridemia, fasting hypoglycemia and postprandial hyperlactatemia. Interestingly, only about 55% of patients showed fasting hypoglycemia, while elevated transaminases were present in nearly all the patients. 

Most patients presented in early childhood, about 75% within the first three years of life. Interestingly, no patient with neonatal presentation has been reported to date, but the youngest patient was reported with recurring feeding difficulties and low borderline blood glucose concentrations as early as at the age of 5 weeks [6]. Manifestation in adulthood is rare, as only one patient with adult presentation has been reported [17]. In this patient, the main clinical symptoms were asthenia, hypoglycemia and fatigue after exercise at age 38. However, it is possible that subtle symptoms were present earlier in this patient. 

There is significant clinical variability between patients. Similar to GSD IX, GSD VI has generally been considered a benign disorder, as most symptoms tend to improve with age. However, of 37 patients who underwent liver biopsy, fibrosis of different degrees was found in about one third and cirrhosis was diagnosed in about 10% of patients as early as at preschool age, in two patients, even within the second year of life. Interestingly, all four patients were Chinese but did not carry the same *PYGL* variants. Ogawa et al. reported a case of GSD VI complicated by focal nodular hyperplasia at age 15 [18]. This patient was not included in our literature review, as the diagnosis was not confirmed genetically. Manzia et al. reported one case of hepatic carcinoma in a 25-year-old patient [19]. The patient received liver transplantation but died about one year later due to neoplastic dissemination and multiorgan failure. Although this patient was classified as having GSD VI by the authors, the information on enzyme analyses suggests that the patient suffered from GSD IX (low levels of phosphorylase-B-kinase in the liver (9.98 IU/g; normal range: 90 +/− 10), muscle (32 IU/g; normal range: 101 +/− 29) and erythrocytes (0 IU/g; normal range: 5 +/− 26.6)) [19]. Genetic results were not available. From the published data, no correlations between early cirrhosis and other disease markers, such as liver transaminases, or the genotype that could be used as predictors of the hepatic phenotype or risk of cirrhosis can be seen. The pathophysiology of hepatic fibrosis and cirrhosis in GSD VI is not fully understood. Wilson at al. have shown in a mouse model that the GSD VI-associated accumulation of hepatic glycogen with age results in liver damage, inflammation and collagen deposition, which can increase the risk of liver fibrosis [20]. While old Pygl−/− mice were at risk of liver fibrosis, there was no evidence of hepatocellular adenoma or carcinoma in spite of excessive glycogen accumulation in the murine model. Additional mechanistic studies are needed to understand the complete metabolic impact of defective glycogen metabolism on the hepatic pathogenesis in GSD VI [20].

Extrahepatic symptoms apart from growth restriction are the exception in GSD VI. Only one case of mild cardiomyopathy according to echocardiography, without clinical symptoms, has been described [15]. A few patients showed mild muscular hypotonia, muscle weakness or developmental impairment, but otherwise, no neurologic symptoms were reported. Notably, axonal neuropathy has been reported in two patients with enzymatically confirmed GSD VI; however, the diagnosis was not confirmed by mutation analysis in these cases, and therefore, they were not included in this study [21]. Recently, the first successful pregnancy in a woman with GSD VI has been reported [16].

About one half of the patients were compound heterozygous for *PYGL* gene variants; the other half carried homozygous variants. In our cohort of 63 patients, a total of 63 *PYGL* variants were detected, including 36 missense mutations, seven stop mutations, 12 splice site variants, seven deletions and one insertion. Our data do not reveal clear genotype–phenotype associations, especially with respect to the more severe phenotypes presenting with cirrhosis. However, this list of known *PYGL* variants is not complete, as several papers with genetic findings in the *PYGL* gene were excluded from this review due to incomplete or lacking clinical information on the affected patients [11,12]. An overview on pathogenic *PYGL* variants reported in the literature, the Human Gene Mutation Database (HGMD) and ClinVar database has recently been published by Liu et al. [22]. 

### 4.1. GSD VI versus GSD IX

One aim of this work was to compare the clinical phenotype of GSD VI to GSD IX, especially since both diseases have many clinical similarities and both present with heterogeneous phenotypes. In the article by Fernandes et al. [9], the age at diagnosis of GSD IX patients was reported to range between 1.81 and 4 years for the different subtypes of the disorder. However, the authors have not defined the term “age at diagnosis”, and it remains unclear if this refers to the age at presentation or specific diagnostic results, such as those from genetic or enzymatic analyses. As the exact age at diagnosis was usually not given in the publications, we decided to report the median age at initial presentation/first symptoms, which was 1.8 years in our GSD VI cohort (range: 5 weeks to 38 years). 

Table 3 gives an overview on the frequency of the clinical and laboratory findings in patients with GSD VI compared to GSD IX. 

The frequency of hepatomegaly in GSD IX (93.2–100% for the different subtypes of GSD IX) was similar to that in GSD VI (96.8%), while growth delay was a more frequent presentation in patients with GSD IX (53.4% for GSD VI; 58.3–87.5% for the different subtypes of GSD IX) [9]. One interesting finding is that developmental delay was a rather common finding, with a prevalence between 25% and 50% for the different subtypes of GSD IX, whereas developmental impairment and delay in motor skills were reported in 6% and 4% of GSD VI patients, respectively. It can be assumed that these neurological impairments were caused by hypoglycemic events. Whether patients with GSD IX have more frequent and/or more severe hypoglycemias cannot be answered based on the published data.

Liver fibrosis/cirrhosis seems to be especially common in GSD IXc (95.8%) patients [9], but also GSD IXa was associated with a higher prevalence of fibrotic or cirrhotic changes compared to GSD VI. However, age at biopsy in our cohort was not reported for all patients. 

Fernandes et al. have shown that GSD IXc patients suffer from more clinically severe liver symptoms and at an earlier age compared to patients with GSD IXa and IXb [9]. One patient with GSD IXc received a liver transplant at the age of 20 years, due to cirrhosis and liver failure. One additional patient with GSD IXc developed hepatocellular carcinoma at 27 years of age and was awaiting liver transplant at the time of the report [9]. In contrast, no patients with genetically confirmed GSD VI, GSD IXa and GSD IXb were reported to have received a liver transplant. However, it must again be stressed that the mean age of the patients is still around pre-school age, and it is possible that patients with a severe liver phenotype may require liver transplantation later in life. 

Overall, the pathogenic reasons for cirrhosis in these special GSD IX subtypes are not fully understood; therefore, it is of great interest that obviously defined, but not all, GSD VI patients also develop cirrhosis at an early age.

### 4.2. Limitations of This Study

One limitation of this review is the small number of published patient reports. Our selection criteria excluded most articles that were not written in English and articles without a genetically confirmed diagnosis of GSD VI. Furthermore, reports with exclusive information on the genotype but otherwise missing or incomplete clinical data were also excluded. Additionally, it is possible that our search strategy might have missed single possible cases. 

As our analysis was not a prospective study but a data analysis based on published information of case reports and case series, there is probably a certain bias, i.e., with respect to the frequency of reported symptoms, laboratory findings, etc. It is well conceivable that symptoms such as mild muscular weakness might have been present but were not reported in the respective publications.

Unfortunately, not much information is available on the long-term outcome of GSD VI. This is due to the fact that many patients were reported soon after their initial presentation and, often, no follow-up data are given. The median age at the time of the report was only 5.3 years. A long-term outcome study on patients with hepatic GSDs including GSD types III, VI and IX followed by the metabolic center in Warsaw, Poland, was recently published [13]. As the diagnoses in this patient cohort were not confirmed genetically, these patients were not included in our review. The study by Szymanska et al. included seven adult patients with GSD VI at a mean age of 23.3 years [13]. The authors did not distinguish between the different forms of ketotic GSDs (type III, VI and IX) in their analysis. Regarding the ketotic GSDs, they found that the most frequent complication was short stature, observed in five out of 16 patients. All but one patient with GSD VI were in balanced metabolic condition at the age of 18 [13].

Published data with confirmed molecular diagnosis and comprehensive clinical information are nonetheless insufficient for drawing conclusions on the effects of dietary treatment. Longitudinal studies are urgently required to better understand the natural history of liver GSD VI, the development and pathophysiology of liver cirrhosis in some cases and the effects of different treatment approaches.

## 5. Conclusions

GSD VI is a disorder presenting with broad clinical heterogeneity. Neither clinical nor laboratory findings allow for a differentiation between GSD VI and GSD IX. Whereas in GSD IX, cirrhosis is most commonly associated with the genetic subtype c, early cirrhosis is found in a small cohort of GSD VI patients, without special clinical or genetic characteristics, and there are no early predictors of severe liver disease. However, it is unknown whether a larger group of patients will develop cirrhosis later on, since the current follow-up time is rather short. Given the low number of published cases, the often benign and unspecific phenotype and the fact that confirmation diagnostics requires either enzymatic or genetic testing, it can be assumed that GSD VI is likely underdiagnosed. Since a significant number of patients show hepatic fibrosis or even cirrhosis at a young age, longitudinal studies are urgently needed to understand the natural history, including the long-term hepatic complications and the effects of dietary treatment. New treatment approaches such as gene therapy, mRNA therapies or small molecules that target glycogen storage within the cells will offer new perspectives for patients in the future and are currently applied in clinical trials in other hepatic GSDs. With these therapies in sight, early diagnosis will become even more essential to prevent liver fibrosis and cirrhosis and other life-limiting symptoms. 

## Figures and Tables

**Figure 1 genes-12-01205-f001:**
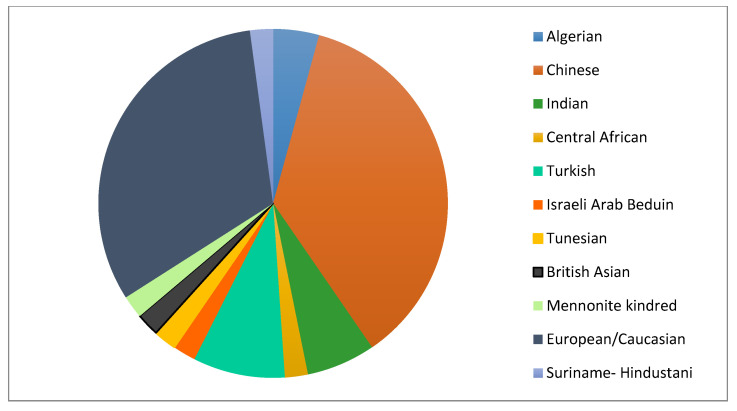
Ethnic/geographic background of 47 families affected with GSD VI.

**Figure 2 genes-12-01205-f002:**
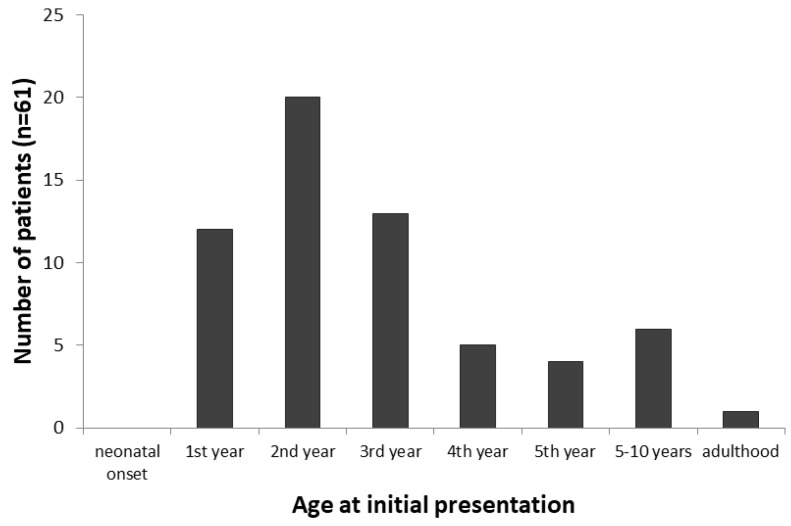
Age at initial presentation of 61 patients with GSD VI. The majority of patients present at pre-school age, while no cases with neonatal and only one case with adult onset were reported.

**Table 1 genes-12-01205-t001:** Clinical findings and their frequencies at initial presentation in patients with GSD VI.

Clinical Finding	Frequency at Initial Presentation
Hepatomegaly	61/63 (96.8%)
Poor growth/short stature	31/58 (53.4%) *
Developmental impairment	3/50 (6.0%) ^§^
Mild delays in motor skills	2/50 (4.0%)
Muscular hypotonia	2/50 (4.0%)
Exercise fatigue	4/50 (8.0%)
Muscle weakness	4/50 (8.0%)
Muscle cramps	4/50 (8.0%)
Splenomegaly	3/63 (4.8%)

* including one patient who was reported to grow slower than his elder brother, but within the normal range. ^§^ One patient with “academic achievement below average”, one patient with “mild delays in the development of speech and motor skills” and one patient with “severe hypoglycemia with convulsions, resulting in developmental delay”.

**Table 2 genes-12-01205-t002:** Laboratory findings, their frequency at initial presentation and range of laboratory values reported in patients with GSD VI.

Laboratory Finding	Frequency at Initial Presentation	Range ^§^
Hypertriglyceridemia	32/53 (60.4%)	normal-15.9 mmol/L
Hypercholesterolemia	12/42 (28.6%)	normal-6.6 mmol/L
Fasting hypoglycemia	29/53 (54.7%)	2.6 mmol/L-normal
Elevated transaminases	54/58 (93.1%)	ASAT 37-829 U/L ALAT 24-1143 U/L
Elevated creatin kinase	0/26 (0%)	all within normal range
Postprandial hyperlactatemia	21/38 (55.3%) *	1.0–8.8 mmol/L

* In many patients postprandial lactic acid elevations were especially reported following an oral glucose tolerance test. ^§^ Exact values were not available from all patients.

**Table 3 genes-12-01205-t003:** Comparison of the frequency of clinical and laboratory findings in patients with GSD VI and GSD IX.

Number of Patients with	GSD VI	GSD IX
Genetically confirmed disease	63	230
Hepatomegaly	61/63 (96.8%)	210/223 (94.2%)
Liver fibrosis	12/37 (32.4%)	36/73 (49.3%)
Liver cirrhosis	4/37 (10.8%)	10/73 (13.7%)
Short stature/poor growth	31/58 (53.4%)	122/200 (61%)
Developmental impairment/delays in motor skills	4/50 (8.0%)	44/120 (36.7%)
Hypertriglyceridemia	32/53 (60.4%)	87/123 (70.7%)
Hypercholesterolemia	12/42 (28.6%)	47/103 (45.6%)
Fasting hypoglycemia	29/53 (54.7%)	79/156 (50.6%)
Elevated transaminases	54/58 (93.1%)	154/169 (91.1%)

## Data Availability

Not applicable.

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
