# Peer review of "The Phenotypic and Genetic Spectrum of Glycogen Storage Disease Type VI"

_genes, 2021, doi:10.3390/genes12081205_

Round 1

Reviewer 1 Report

In this manuscript, Grünert et al. performed a systematic literature review of published cases of patients with glycogen storage disease type VI (GSD VI). Most of the data is extracted from the recent literature. This article is built on the same analysis scheme as the one proposed by Fernandes et al. (2020), who described the symptoms related to the different subtypes of GSDIX. This allowed the authors to compare GSDVI and GSDIX diseases

-          First, it would be interesting to add the extreme values (or the range values) of the studied parameters in Table 2. For example, add a column to indicate the range values of ASAT/ALAT, blood glucose, ketones, TG probably detailed in the original articles.

-          As GSDVI develop hypoglycemia, does the term fasting indicate a measurement of blood sugar after a night of fasting? Please add details.

-          To easily compare GSDVI and GSDIX, could the authors add a comparative Table in the discussion section.

-          What is the life expectancy of these patients? This point can be discussed according to the nutritional management and the good metabolic balance of the patients

-          Can growth delay be linked to the frequency of hypoglycemic episodes?

-          Why hypoglycemia is not mentioned in the main symptoms. Explain what is happening metabolically to endogenous glucose production in relationship to glycogenolysis.

-          I don't agree with this sentence: “However, age at biopsy in our cohort was not reported for all patients, and the mean age at biopsy was younger than in the GSD IX cohort (mean age at biopsy 2.9 years in GSD VI versus 3.03-3.5 years GSD IXa-c), what may also have an effect on the histological findings»

-          The authors mentioned some data obtained in a mouse model of GSD VI, do these mice have extrahepatic symptoms

-          I would suggest rewriting the discussion avoiding re-describing the results (lots of repetition) and focusing only on the comparison between GSDVI and GSDIX. It would be also interesting to compare and mention GSD III phenotype.

Minor comments:

-          In the abstract, please mention phosphorylase kinase deficiency for GSDIX to emphasize the interest to compare these GSD diseases.

-          I would like to suggest presenting the results of the ethnic distribution (line 125) as a pie chart.

-          Line 77, please change GSD6 in GSD VI.

-         What is the histological grade of fibrosis mentioned at line 156?

Author Response

Dear Editors,

We thank the reviewers for their valuable comments on our manuscript “The phenotypic and genetic spectrum of glycogen storage disease type VI”.

This is how we addressed the reviewers’ comments:

Reviewer #1:

In this manuscript, Grünert et al. performed a systematic literature review of published cases of patients with glycogen storage disease type VI (GSD VI). Most of the data is extracted from the recent literature. This article is built on the same analysis scheme as the one proposed by Fernandes et al. (2020), who described the symptoms related to the different subtypes of GSDIX. This allowed the authors to compare GSDVI and GSDIX diseases

-          First, it would be interesting to add the extreme values (or the range values) of the studied parameters in Table 2. For example, add a column to indicate the range values of ASAT/ALAT, blood glucose, ketones, TG probably detailed in the original articles.

            We agree with the reviewer and have added the requested information to Table 2.

-          As GSDVI develop hypoglycemia, does the term fasting indicate a measurement of blood sugar after a night of fasting? Please add details.

This is often not clearly stated in the respective case reports. Therefore, we cannot add any reliable information here.

-          To easily compare GSDVI and GSDIX, could the authors add a comparative Table in the discussion section.

According to the reviewer’s suggestion we have added a comparative table to the discussion section.

-          What is the life expectancy of these patients? This point can be discussed according to the nutritional management and the good metabolic balance of the patients.

Unfortunately, we cannot draw any conclusions on life expectancy from our dataset as most patients are reported in childhood. It can be assumed that life expectancy is probably not much impaired if no hepatic complications occur, however, this is only hypothetic.

-          Can growth delay be linked to the frequency of hypoglycemic episodes?

The frequency of hypoglycemic episodes is only reported in very few articles. Therefore, no reliable conclusions can be drawn from the published cases.

-          Why hypoglycemia is not mentioned in the main symptoms. Explain what is happening metabolically to endogenous glucose production in relationship to glycogenolysis.

In the Results section, hypoglycemia is not in the list of main symptoms as we characterized it as a laboratory finding. It is mentioned as such in the text as well as in Table 2.

It is additionally mentioned in the Introduction as a main feature of the disease: “Clinical and laboratory findings comprise hepatomegaly, poor growth and short stature, ketotic hypoglycemia, elevated hepatic transaminases, hypertriglyceridemia and hypercholesterolemia.”

-          I don't agree with this sentence: “However, age at biopsy in our cohort was not reported for all patients, and the mean age at biopsy was younger than in the GSD IX cohort (mean age at biopsy 2.9 years in GSD VI versus 3.03-3.5 years GSD IXa-c), what may also have an effect on the histological findings»

We agree with the reviewer. This was a typo and must be “age at biopsy was younger than in the GSD VI cohort”. This has been corrected. As Fernandes et al. have used the mean age at biopsy and not the median age, we have additionally added the mean age of the GSD VI cohort to the Results section.

-          The authors mentioned some data obtained in a mouse model of GSD VI, do these mice have extrahepatic symptoms

No, as in GSD VI patients extrahepatic symptoms have not been reported in the mouse model. The phenotype very much resembles the human phenotype: “Pygl-deficient mice exhibit hepatomegaly, excessive hepatic glycogen accumulation, mild fasting hypoglycemia, and elevated blood ketone bodies during prolonged fasting. Furthermore, we show PYGL deficiency leads to progressive accumulation of hepatic glycogen with age and increases the risk of liver damage and inflammation along with collagen deposition in old Pygl −/− mice. (Wilson et al. 2020)

-          I would suggest rewriting the discussion avoiding re-describing the results (lots of repetition) and focusing only on the comparison between GSDVI and GSDIX. It would be also interesting to compare and mention GSD III phenotype.

We thank the reviewer for this suggestion, however, we think a comparison with GSD III would require a similarly intensive and comprehensive literature review on GSD III as done for GSD VI and IX, which is beyond the scope of this article. However, this could be a future project.

Minor comments:

-          In the abstract, please mention phosphorylase kinase deficiency for GSDIX to emphasize the interest to compare these GSD diseases.

            We have added this information to the abstract.

-          I would like to suggest presenting the results of the ethnic distribution (line 125) as a pie chart.

            We have added a pie chart on the ethnic/geographic background of the patients.

-          Line 77, please change GSD6 in GSD VI.

            This has been changed.

-         What is the histological grade of fibrosis mentioned at line 156?

Unfortunately, the manuscript document that can be downloaded from the website has no line numbering. However, if the grade of fibrosis was not mentioned in the text, it was probably not specified in the original case report.

Reviewer #2:

This work is a literature review collecting clinical data and performing descriptive statistical analysis on the glycogen storage disease type VI. After that, there is a comparison of the clinical phenotype of the glycogen storage disease type IX finding an interesting developmental delay distinction between the two mutational disease. It is well organized and the authors discuss right about the limitations of this review linked to the diagnosis leak of this disease. This paper is interesting for clinical reflections about this kind of diseases.

Thank you for this reviewing work, it is pretty clear and interesting. 

We thank reviewer #2 for his very positive evaluation of our work. No revisions have been requested.

With many thanks and best regards,

PD Dr. med. Sarah. C. Grünert

Reviewer 2 Report

This work is a literature review collecting clinical data and performing descriptive statistical analysis on the glycogen storage disease type VI. After that, there is a comparison of the clinical phenotype of the glycogen storage disease type IX finding an interesting developmental delay distinction between the two mutational disease. It is well organized and the authors discuss right about the limitations of this review linked to the diagnosis leak of this disease. This paper is interesting for clinical reflections about this kind of diseases.

Thank you for this reviewing work, it is pretty clear and interesting. 

Author Response

We thank reviewer #2 for his very positive evaluation of our work. No revisions have been requested.